# A Risk Assessment for Ozone Regulation Based on Statistical Rollback

Yongku Kim[1,*] and Jeongjin Lee [2]

1    Department of Statistics, Kyungpook National University, Daegu 41566, Korea
2    Department of Statistics, Colorado State University, Fort Collins, CO 80523, USA; stecophil88@gmail.com
*    Correspondence: kim.1252@knu.ac.kr

**Abstract:** In environmental studies, it is important to assess how regulatory standards for air pollutants affect public health. High ozone levels contribute to harmful air pollutants. The EPA regulates ozone levels by setting ozone standards to protect public health. It is thus crucial to assess how various regulatory ozone standards affect non-accidental mortality related to respiratory deaths during the ozone season. The original rollback approach provides an adjusted ozone process under a new regulation scenario in a deterministic fashion. Herein, we consider a statistical rollback approach to allow for uncertainty in the rollback procedure by adopting the quantile matching method so that it provides flexible rollback sets. Hierarchical Bayesian models are used to predict the potential effects of different ozone standards on human health. We apply the method to epidemiologic data.

**Keywords:** hierarchical model; mortality; ozone regulatory standard; risk assessment; stochastic rollback

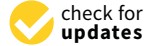

## 1. Introduction

Regulating high ozone levels is essential as exposure to ozone can affect the risk of respiratory diseases or related deaths. Tropospheric ozone, also known as ground-level ozone, is one of the main harmful air pollutants that can cause adverse health effects. Many areas in the United States have been observed to exceed the current ozone National Ambient Air Quality Standards (NAAQS). Elevated ozone concentrations have also been a growing concern for rapidly developing nations where emissions of ozone precursors have been risen from expanding transportation networks [1]. Recently, U.S. EPA promulgated a new ozone NAAQS as 0.070 ppm (parts per million) and this change has prompted studies on the effects of the new ozone standard [2–4]. To investigate how different ozone NAAQS affect public health and welfare can be of great interest in epidemiologic studies, toxicological studies, or controlled human exposure studies. Based on the reviews of the air quality criteria for ozone ($O_3$) and related photochemical oxidants and the NAAQS for $O_3$, a modification in the current ozone regulatory standards provides the required protection for public health and welfare (see also [5,6]). Bell et al. [7,8] disapproved of the EPA's scientific reviews reporting that designations based on air quality data from 2006 to 2008 would be effective in air quality data to take effect in 2010 for the 2007 8-h ozone standard. Therefore, it would be interesting to investigate whether the current regulations are sufficiently stringent to prevent respiratory-related mortality or not.

To assess how changes in the ozone regulations affect mortality, ground-level ozone must be adjusted by strengthening the air quality standards. Although rollback functions, namely, air quality adjustment procedures proposed by the EPA [9], can be useful for adjusting the ozone process, it cannot introduce sufficient variability in the rollback adjustment as the adjustment is deterministic and the EPA regulatory standards are based on the average of three consecutive years' AQI values. Thus, we consider a parametric rollback approach with quantile matching method to allow for uncertainty in the rollback procedure

To conduct the risk assessment, we consider hierarchical Bayesian models that provide uncertainty quantification for relevant parameters to predict the potential effects of different regulatory ozone standards. We also describe some variations in the results under different modeling assumptions. We analyze databases from the National Morbidity, Mortality, and Air Pollution Study [10] that is designed to study the public health effects of air pollutants. The Health Effects Institutes began this study with researchers from Johns Hopkins and Harvard University in 1996. The NMMAPS database contains 108 U.S. urban areas from 1987 to 2000 where one can build a multisite time-series model of ozone and mortality simultaneously with meteorology information such as temperature or dew point and air pollution ($O_3$, $PM_{10}$, $SO_2$, $NO_2$, and CO) (see also [11]).

The focus of this paper is on statistical approaches to describe how the new ozone regulations affect mortality. We shall describe our results that mortality decreases as limits of acceptable ozone level get lower through the statistical rollback approach.

## 2. Statistical Rollback Approach

The rollback transformation [9] is one of the methods to adjust current ozone processes to follow new ozone regulatory standards. There are two main conditions to consider in air quality data. The first one is baseline conditions characterized by unadjusted air quality data monitored at fixed locations during recent years. The other one is attainment conditions generated by fitting an air quality adjustment procedure (AQAP) to the baseline. QQ-plots of high ozone levels in 1992 under baseline conditions against low ozone levels in 1999 under attainment conditions are shown in Figure 1. The figure exhibits different behaviors at high quantiles.

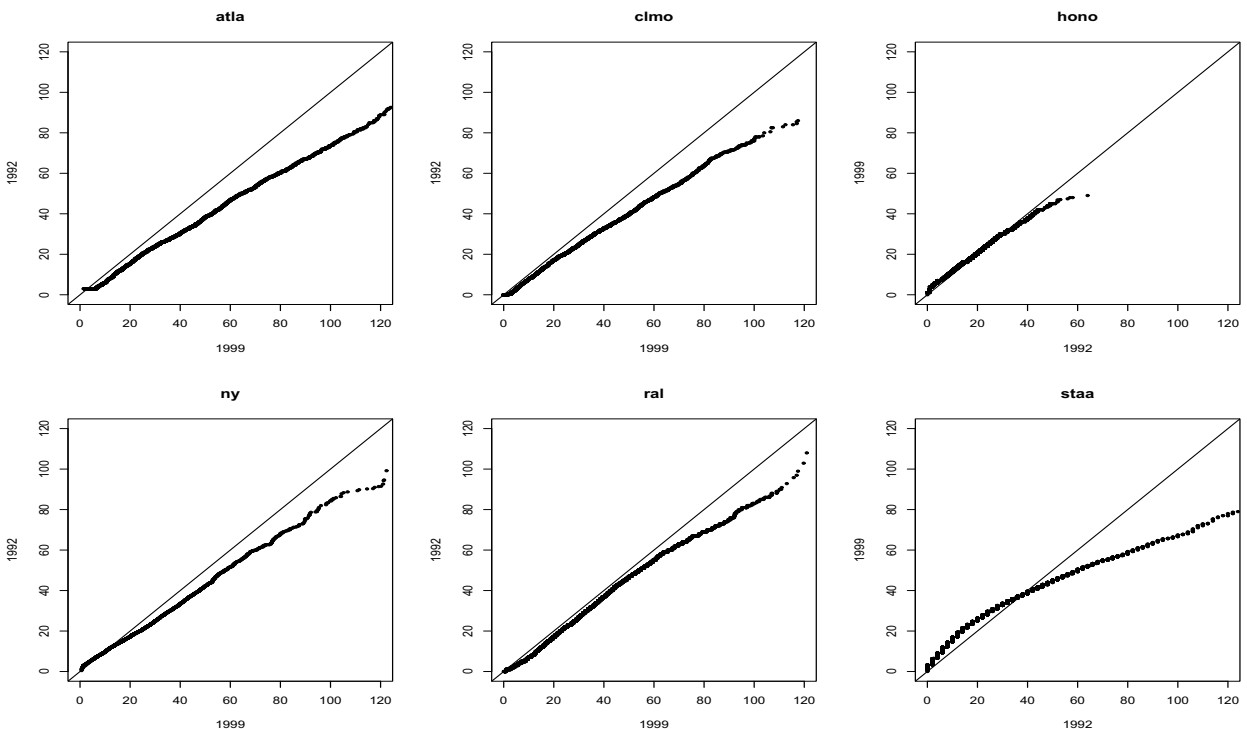

**Figure 1.** QQ-plots of high ozone levels in 1992 under baseline conditions against low ozone levels in 1999 under attainment conditions at six different locations.

### 2.1. Quantile Matching Approach

Suppose that current ozone level, denoted by $x$, is a random variable having a distribution function $F_{\theta}$. We assume that adjusted ozone level corresponding to $x$ under a new scenario follows the same distribution with different parameters, $F_{\theta^*}$, and there exists a mapping $m$ such that a random variable $m(x)$ has the distribution function $F_{\theta^*}$. Thus,

the adjusted ozone level can be obtained using $m(x)$. That is, $\boldsymbol{\theta}^*$ is the adjusted model parameter based on the relationship between current levels and adjusted levels under the new scenario. Moreover, let $F(x; \boldsymbol{\theta})$ and $G(z; \boldsymbol{\gamma})$ be the ozone distribution functions based on the baseline condition and attainment condition, respectively, (e.g., $G \prec F$). That is, $m(x) = F_{\boldsymbol{\theta}^*}^{-1}(F_{\boldsymbol{\theta}}(x))$ or $m(x) = G^{-1}(F(x))$. Note that $F$ and $G$ can be estimated by baseline and attainment years' ozone data, respectively. Attainment year based on its AQI values or the condition $G \prec F$ is required. $G$ may require adjustment based on $\rho(s) = \frac{\mathbf{x}(b)}{\mathbf{x}(s)}$, where $\mathbf{x}(b)$ is current design values and $\mathbf{x}(s)$ is design values under scenario $s$.

Let $f_{\boldsymbol{\theta}}$ be a density function of current ozone levels $x$. Then adjusted rollback values $z$ corresponding to $x$ are obtained as follows:

1.  Based on current ozone levels $\mathbf{x} = \{x_i\}_{i=1}^n$, estimate parameters $\boldsymbol{\theta}$ (i.e., $\hat{\boldsymbol{\theta}}$).
2.  Compute $q_i^x$, the quantile of $x_i$ such that

$$q_i^x = \int_{-\infty}^{x_i} f_{\hat{\boldsymbol{\theta}}}(z)dz,$$

for $i = 1, \ldots, n$.
3.  Estimate parameters $\boldsymbol{\theta}^*$ under new scenario (i.e., $\hat{\boldsymbol{\theta}}^*$).
4.  Determine the corresponding $z_i$ satisfying

$$q_i^x = \int_{-\infty}^{z_i} f_{\hat{\boldsymbol{\theta}}^*}(z)dz,$$

for $i = 1, \ldots, n$.
5.  $\{z_i\}_{i=1}^n$ are adjusted (rollback) values of $\mathbf{x}$.

We extend this approach through the Bayesian framework by putting prior distributions of parameters, $\pi(\boldsymbol{\theta})$. By generating samples from posterior distribution, $\pi(\boldsymbol{\theta}|\mathbf{x}) \propto f_{\boldsymbol{\theta}} \pi(\boldsymbol{\theta})$, we can generate rollback ensembles in the same fashion.

In general, the distribution of $x$ can be expressed as a mixture of the form

$$f(x) = \begin{cases} (1-w)h(x|\boldsymbol{\theta}_1) & x < u \\ wg(x|\boldsymbol{\theta}_2) & x \geq u \end{cases},$$

where $u$ is a threshold, $h$ can be any parametric distribution such as Weibull, truncated Gamma or Gaussian distribution, and $g$ is a Generalized Pareto distribution (GPD) with parameters $\boldsymbol{\theta}_2 = (\sigma, \xi)$.

The quantile matching approach is simple and straightforward but causes various problems in adjusting multiple ozone processes in large dimensions. Now we consider more feasible approaches in the next section.

### 2.2. Weibull Approach in Rollback

Let $y_t(c)$ denote hourly (or daily average, daily maximum, daily 8-h maximum) ozone level for each time $t$ and city $c$ and $y_t^*(c)$ denote rollbacked process corresponding to baseline process $y_t(c)$. That is,

$$y_t^* = \gamma(y_t)^{\eta},$$

where we omit city index for simplicity. We fit $y_t$ to a parametric distribution such as Weibull distribution.

We obtain following information for each year $i$ and each city $c$:

$$(AQI_i(c), \kappa_i(c), \delta_i(c)),$$

where $\kappa$ and $\delta$ are weibull parameters. Note that $\gamma$ and $\eta$ are functions of $\kappa$ and $\delta$.

Now we consider a model of the form

$$E\begin{pmatrix} \kappa_i(c) \\ \delta_i(c) \end{pmatrix} = \begin{bmatrix} g_1(AQI_i(c); \boldsymbol{\theta}_1) \\ g_2(AQI_i(c); \boldsymbol{\theta}_2) \end{bmatrix} \text{ and } Cov\begin{pmatrix} \kappa_i(c) \\ \delta_i(c) \end{pmatrix} = \Sigma(c).$$

For example, we consider a simple linear model:

$$E\begin{pmatrix} \kappa_i(c) \\ \delta_i(c) \end{pmatrix} = \begin{bmatrix} 1 & AQI_i(c) & 0 & 0 \\ 0 & 0 & 1 & AQI_i(c) \end{bmatrix} \begin{bmatrix} \alpha_1(c) \\ \beta_1(c) \\ \alpha_2(c) \\ \beta_2(c) \end{bmatrix}$$

and

$$Cov\begin{pmatrix} \kappa_i(c) \\ \delta_i(c) \end{pmatrix} = \Sigma(c) = \begin{bmatrix} \sigma_{11}(c) & \sigma_{12}(c) \\ \sigma_{21}(c) & \sigma_{22}(c) \end{bmatrix}.$$

For new standard regulation $AQI^*$, we can obtain $\kappa^*(c)$ and $\delta^*(c)$ for each city $c$ using

$$\kappa^*(c) = \alpha_1(c) + \beta_1(c)AQI^* \text{ and } \delta^*(c) = \alpha_2(c) + \beta_2(c)AQI^*.$$

It is observed that $\Sigma(c)$ introduces some uncertainty in $\kappa^*(c)$ and $\delta^*(c)$. In general, hierarchical structures for $(\alpha_1(c), \beta_1(c), \alpha_2(c), \beta_2(c))$ can be considered. Usually, $\delta$ provides a better fit based on $ACLV1$. Thus, we may need to adopt the relationship between $\kappa$ and $\delta$:

$$\delta = \frac{ACLV1}{\log(n)^{1/\kappa}},$$

where

$$ACLV1 = TAQI \times \frac{CLV1}{AQI}.$$

Otherwise, we may select an average value or baseline value of $\kappa$ or $\delta$.

Let $PAQI_s(c)$ be the 4th largest 8-h daily maximum concentration of $z_t$, which is

$$\delta^*\left(\frac{x_t}{\delta}\right)^{\kappa/\kappa^*}.$$

Then,

$$y_t^* = \frac{TAQI_a(c)}{PAQI_s(c)}z_t.$$

*2.3. Log-Normal Approach in Rollback*

Let $y_t$ be a baseline process and $y_t^*$ be rollbacked process. Assume that

$$y_t^* = \gamma(y_t)^\eta,$$

where $y_t \sim logNormal(\mu, \sigma^2)$. That is,

$$\log y_t \sim N(\mu, \sigma^2) \text{ and } \log y_t^* \sim N(\lambda + \eta\mu, \eta^2\sigma^2),$$

where $\lambda = \log \gamma$.

Similarly, we fit $y_t$ to a log-normal distribution for each time $t$ and city $c$. It is observed that $y_t$ can be any covariate in the risk model (e.g., daily average, daily maximum, and daily 8-h maximum). Then, can obtain following information for each year $i$ and each city $c$:

$$(AQI_i(c), \mu_i(c), \sigma_i(c)).$$

Now we consider a simple linear model:

$$E\begin{pmatrix} \mu_i(c) \\ \sigma_i(c) \end{pmatrix} = \begin{bmatrix} 1 & AQI_i(c) & 0 & 0 \\ 0 & 0 & 1 & AQI_i(c) \end{bmatrix} \begin{bmatrix} \alpha_3(c) \\ \beta_3(c) \\ \alpha_4(c) \\ \beta_4(c) \end{bmatrix}$$

and

$$Cov\begin{pmatrix} \mu_i(c) \\ \sigma_i(c) \end{pmatrix} = \Sigma(c) = \begin{bmatrix} \sigma_{33}(c) & \sigma_{34}(c) \\ \sigma_{43}(c) & \sigma_{44}(c) \end{bmatrix}.$$

For new standard regulation $AQI^*$, we can determine $\mu^*(c)$ and $\sigma^*(c)$ for each city $c$ using

$$\lambda + \eta\mu = \alpha_3(c) + \beta_3(c)AQI^* \quad \text{and} \quad \eta\sigma = \alpha_4(c) + \beta_4(c)AQI^*,$$

where $\mu$ and $\sigma$ are estimated based on baseline year. In general, the following relationship is considered:

$$\log\frac{TAQI}{AQI} + \mu + \sigma\Phi^{-1}\left(1 - n^{-1}\right) = \lambda + \eta\mu + \eta\sigma\Phi^{-1}\left(1 - n^{-1}\right).$$

It is observed that $\eta = \kappa/\kappa^*$ and $\lambda = \log\gamma = \log(\delta^*) - \frac{\kappa}{\kappa^*}\log(\delta) = \log(\delta^*) - \eta\log(\delta)$. Furthermore, well-fitted models of $\mu(c)$ and $\delta(c)$ can provide a good estimation based on following relationships

$$\lambda + \eta\mu = \alpha_3(c) + \beta_3(c)AQI^* \quad \text{and} \quad \lambda = \log(\delta^*) - \eta\log(\delta).$$

That is,

$$\hat{\eta} = \frac{\alpha_3(c) + \beta_3(c)AQI^* - \log(\delta^*)}{\mu - \log(\delta)}$$

and

$$\hat{\lambda} = \log(\delta^*) - \hat{\eta}\log(\delta).$$

We can estimate one parameter (e.g., $\delta^*$ or $\mu^*$) first and then tune other parameter (e.g., $\kappa^*$ or $\sigma^*$) to make rollbacked AQI set to be new standard regulation $AQI^*$ based on the range of the parameter. In the 8-h daily maximum rollback, we can directly solve the following equation by plugging in estimated parameters of $\delta^*$ or $\mu^*$:

$$AQI^* = \gamma(AQI)^{\eta},$$

where $\eta = \kappa/\kappa^*$ and $\gamma = \delta^*/\delta^{\frac{\kappa}{\kappa^*}}$. That is,

$$\hat{\kappa}^* = \frac{\kappa\log\frac{AQI}{\delta}}{\log\frac{AQI^*}{\delta^*}}.$$

Furthermore, the following empirical relationships can be useful.

$$\text{daily 1-h maximum} = 2.5 \times \text{daily average}$$

and

$$\text{daily 8-h maximum} = 1.33 \times \text{daily average}.$$

As previous approaches (except for the 8-h maximum covariate) do not guarantee rollbacked values satisfying exact new standard regulation $AQI^*$, hourly scale rollback will be more appropriate for adjustment after rollback. Notice that EPA standard regulation is based on the average of three consecutive years' AQI values.

## 3. Application to NMMAPS Data

### 3.1. Statistical Modeling

We apply an overdispersed Poisson model in generalized linear models (see [8,12]) to the NMMAPS data. Denote $Y_t^c$ by the number of daily non-accidental deaths in community $c$ on day $t$. The Poisson process with intensity function $\mu_t^c$ can be expressed as

$$Y_t^c \sim Poisson(\mu_t^c) \text{ with } Var(Y_t^c) = \phi_c \mu_t^c, \tag{1}$$

where the parameter $\phi_c$ describes overdispersion for community $c$. Note that all overdispersed Poisson models are assumed to be mutually independent over time. We can also model the intensity function $\mu_t^c$ with some essential covariates such as ozone levels at different lags, seasonality, long-term trends, weather, and co-pollutants for three age groups (<65, 65–74, and ≥75 year). Natural cubic splines are useful tools for getting smoothing functions of time to account for seasonality and long-term trends in which influenza epidemics, for example, can affect mortality. The interaction term between smoothing functions of time and age-specific indicators (<65, 65–74, and ≥75 years) is considered as it can adjust the possible seasonal mortality patterns by age group. We also control for some potential covariates related to weather by smoothing functions of dew point, average dew points of the previous three days, temperature, the average temperature of the previous three days as follows:

$$\log \mu_t^c = \beta^c x_t^c + \alpha^c DOW_t + \gamma_1^c ns(time, 7/year) + \gamma_2^c ns(T_t^c, 6) \tag{2}$$

$$+ \gamma_3^c ns(T_{t-1,t-3}^c, 6) + \gamma_4^c ns(D_t^c, 3) + \gamma_5^c ns(D_{t-1,t-3}^c, 3) \tag{3}$$

$$+ \text{interaction terms for age groups and time,} \tag{4}$$

where $\mu_t^c$ is the expected number of deaths, $x_t^c$ is the average daily $O_3$ concentrations of the current and the previous day in community $c$ on the day $t$, $DOW_t$ is the days of the week (categorical) on day $t$. We define $ns(time, 7/year)$, $ns(T_t^c, 6)$, $ns(T_{t-1,t-3}^c, 6)$, $ns(D_t^c, 3)$ and $ns(D_{t-1,t-3}^c, 3)$ as the natural cubic spline function of time with 7 degrees of freedom per year, temperature with 6 degrees of freedom, the average temperature of the previous three days with 6 degrees of freedom, dew point with 3 degrees of freedom and the average dew points of the previous three days with 3 degree of freedom, respectively. The last term in (4) indicates the interaction terms of age-specific indicators and natural cubic spline functions of time.

Relative mortality rates associated with exposure to ozone over the past few days can be estimated in a specific community by constrained or unconstrained distributed-lag models as daily ozone levels are readily available. Those models can be more flexible in the sense that they can be more suitable for exploring the time lag between exposures to ozone and deaths than single-lag models (see [7]).

For example, the constrained distributed-lag models (CDL) and unconstrained distributed-lag models (UDL) can be expressed as

$$\beta^c \mathbf{x}_t^c = \beta_0^c x_t + \beta_1^c \bar{x}_{t:t-3}^c + \beta_2^c \bar{x}_{t:t-6}^c \quad \text{or} \quad \beta^c \mathbf{x}_t^c = \sum_{j=0}^{6} \beta_j^c x_{t-j}^c, \tag{5}$$

respectively.

The national average relative mortality rate caused by ozone effects can be estimated through a Bayesian hierarchical model in which both variability within-community and across-community can be accounted. The national average relative rate will be estimated by integrating estimates for the relative rate from the distributed-lag models for each specific community. Through the two-stage model, first, variation across communities is

considered over the short-term ozone effects. The national average relative rate is then estimated. The ozone effects $O_3$ on mortality is modeled for each community $c$ as follows:

$$\hat{\boldsymbol{\beta}}^c | \boldsymbol{\beta}^c, \hat{\Sigma}^c \sim MVN(\boldsymbol{\beta}^c, \hat{\Sigma}^c), \tag{6}$$

where $\boldsymbol{\beta}^c$ is the true relative rate in a specific community, $\hat{\boldsymbol{\beta}}^c$ is its estimate, and $\hat{\Sigma}^c$ is the estimated covariance matrix corresponding to $\hat{\boldsymbol{\beta}}^c$. We put the multivariate normal distribution on $\boldsymbol{\beta}^c$,

$$\boldsymbol{\beta}^c | \boldsymbol{\mu}, \Omega \sim MVN(\boldsymbol{\mu}, \Omega), \tag{7}$$

where $\boldsymbol{\mu}$ is the true national average relative rate and $\Omega$ is the true covariance matrix of relative rate in a specific community, $\boldsymbol{\beta}^c$. Note that there are still several sensitivity issues related to the modeling: (1) co-pollutant such as $PM_{10}$ can be included as a potential confounder; (2) days with high temperatures can be excluded to control for the effects of heat waves; (3) the degrees of freedom (df) in the smooth functions of time needs to be specified to control for long-term trends and seasonality; and (4) various ozone exposure metrics such as daily average, 1-h maximum, and 8-h maximum can be considered (see [7]).

*3.2. Inferences*

Estimating the impact of new ozone regulatory standards on the total nonaccidental deaths is not achieved through the national average relative rate $\boldsymbol{\mu}$ as the rollback adjustment is applied to each community separately. Instead, we directly calculate the expected total nonaccidental deaths of the original and adjusted (rollbacked) ozone process, respectively.

Denote $g_c$ by a rollback function for community $c$, which adjusts observed ozone concentrations to meet new ozone regulation standards. Our focus is then on the difference in expected nonaccidental deaths between before and after rollback transformation. Let $E[\log(\mu_t^c)] = \mathbf{x}_t^c \hat{\boldsymbol{\beta}}^c + \mathbf{M}_t^c \hat{\boldsymbol{\theta}}^c$, where $\mathbf{M}_t^c$ is the design matrix of covariates except for ozone (i.e., $DOW_t$, $ns(time, 7/year)$, $ns(T_t^c, 6)$, and $ns(D_t^c, 6)$, etc.), and $\mathbf{x}_t^c$ is the design matrix of unconstrained (or constrained) distributed-lag ozones for community $c$ (see [7]). If the amount of ozone reduction $r$ is the same for each community $c$, then the reduction ratio of the expected total death is $1 - \exp\left(-\mathbf{r}\hat{\boldsymbol{\beta}}^c\right)$ for each community $c$ at time $t$. Thus, the reduction rate of the national expected total death is $1 - \exp(-\mathbf{r}\hat{\boldsymbol{\mu}})$ at time $t$. As the $\boldsymbol{\mu}$ is obscure, the reduction rate of the national expected total death can be better estimated by weighting mean effects across the 98 cities, where weights are proportional to populations.

For simplicity, the unconstrained distributed-lag model is only considered here although the constrained distributed-lag models can be also readily applied via reparameterization. Let $z_t$ be rollbacked ozone concentrations of $x_t$ for community $c$ (i.e., $z_t^c = g_c(x_t^c)$). After the rollback transformation, the expected total deaths for community $c$ during the ozone season is as follows

$$\sum_{t \in T_{O_3}} \exp\left(\sum_{j=0}^{6} \beta_j g_c(x_{t-j}^c) + \mathbf{M}_t^c \hat{\boldsymbol{\theta}}^c\right), \tag{8}$$

where $T_{O_3}$ is the set of time indices for the ozone season. The log ratio of national expected death before and after the rollback transformation during ozone season can be expressed as

$$\log \sum_{c \in C} \sum_{t \in T_{O_3}} w_{c,t} \exp\left(\sum_{j=0}^{6} \beta_j (z_{t-j}^c - x_{t-j}^c)\right), \tag{9}$$

where $C$ is the set of indices of communities in NMMAPS data and $w_{c,t}$ is the weight of community $c$ at time $t$,

$$w_{c,t} = \frac{\exp\left(\sum_{j=0}^{6} \beta_j x_{t-j}^c + \mathbf{M}_t^c \hat{\theta}^c\right)}{\sum_{c \in C} \sum_{t \in T_{O_3}} \exp\left(\sum_{j=0}^{6} \beta_j x_{t-j}^c + \mathbf{M}_t^c \hat{\theta}^c\right)}. \tag{10}$$

We can estimate the posterior distribution of the log ratio in (9) through the MCMC approach. As $\pi(\boldsymbol{\beta}|\mathbf{Y})$ can be approximated by $\pi(\boldsymbol{\beta}|\hat{\boldsymbol{\beta}})$ and a prior distribution of $(\boldsymbol{\mu}, \Omega)$ well, we can generate $\boldsymbol{\beta}$ from the posterior distribution $\pi(\boldsymbol{\beta}|\mathbf{Y})$. One of the efficient algorithms in application is TLNISE (two level Normal independent sampling estimation) algorithm (see [13]). Without a Bayesian approach, restricted maximum likelihood (REML) method can provide similar estimates.

Finally, with samples from $\pi(\boldsymbol{\beta}|\mathbf{Y})$ the posterior distribution of the log ratio of national expected death in (9) can be estimated as follows

$$\log \frac{\sum_{c \in C} \sum_{t \in T_{O_3}} \exp\left(\sum_{j=0}^{6} \beta_j^{(i)} z_{t-j}^c + \mathbf{M}_t^c \hat{\theta}^c\right)}{\sum_{c \in C} \sum_{t \in T_{O_3}} \exp\left(\sum_{j=0}^{6} \beta_j^{(i)} x_{t-j}^c + \mathbf{M}_t^c \hat{\theta}^c\right)}, \tag{11}$$

where $\boldsymbol{\beta}^{(i)}$ indicates each sample from $\pi(\boldsymbol{\beta}|\mathbf{Y})$ or $\pi(\boldsymbol{\beta}|\hat{\boldsymbol{\beta}})$ for $i = 1, \dots, N$.

## 4. Results

We investigate how the rollback transformation predicts that ozone series change under new ozone regulation standards possibly affecting public health. The transformation is based on the current AQI and new regulation standards in the attainment years. Figure 2 shows Q-Q plots of ozone levles under new regulation standards against the current AQI using various rallback functions and statistical rollbacks. A common AQI is applied to all cities (common rollback) or a different AQI is applied to each city (city-specific rollback).

The regular meteorological model in the NMMAPS is based on nonlinear functions mainly consisting of temperature and dewpoint at lag 0 and the average of them at lags 13. However, meteorological confounding in a distributed-lag model for ozone may exist at lags 46 so that we account for temperature and dewpoint at lag 0 through 6 in the "distributed-lag" meteorological model through nonlinear splines with 4 and 3 df, respectively.

We use the reduction rate of the expected total death to assess the effect of the proposed new ozone regulatory. Posterior means and 95% credible intervals for total mortality per 1000 deaths are shown in Tables 1 and 2. They are based on unconstrained distributed-lag models with a common rollback and city-specific rollback approach, respectively. Most credible intervals have positive lower limits in the common rollback approach except for Weibull rollback at level 75 regularization, providing good evidence that there are reduced rates. As expected, the mortality rate decreases as lower regulation increases. Similar results are observed in CDL. The ranges of the estimated reduction rates are approximately 0.94–1.9 for 75 ppb, 1.47–2.9 for 70 ppb and 4.9–5.0 for 60 ppb for all models and rollback functions. Statistical rollbacks tend to show similar results as quadratic rollbacks for overall regularization levels. The ranges of the estimated reduction rates are roughly 1.0–2.0 for 75 ppb, 1.4–2.8 for 70 ppb and 2.4–4.8 for 60 ppb for all rollback functions in the city-specific rollback approach. Posterior variances of the relative risks for each specific city vary with rollback functions. Rather than we include daily average ozone concentration as a covariate, we include daily maximum ozone concentration and daily 8-h maximum ozone concentration so that they reduce the mortality rate more in Table 3. We also conduct statistical inference with simple MLE and pooled MLE and then compare them with Bayesian inference. Both MLE and pooled MLE give a slightly higher reduction rate than one based on the Bayesian inference in Table 4. Several issues associated with high

temperature on the mortality reduction. These are investigated by presenting mortality reduction with and without high temperaturein Table 5.

Several issues associated with high temperature on the mortality reduction. These are investigated by presenting mortality reduction with and without high temperature in Table 5.

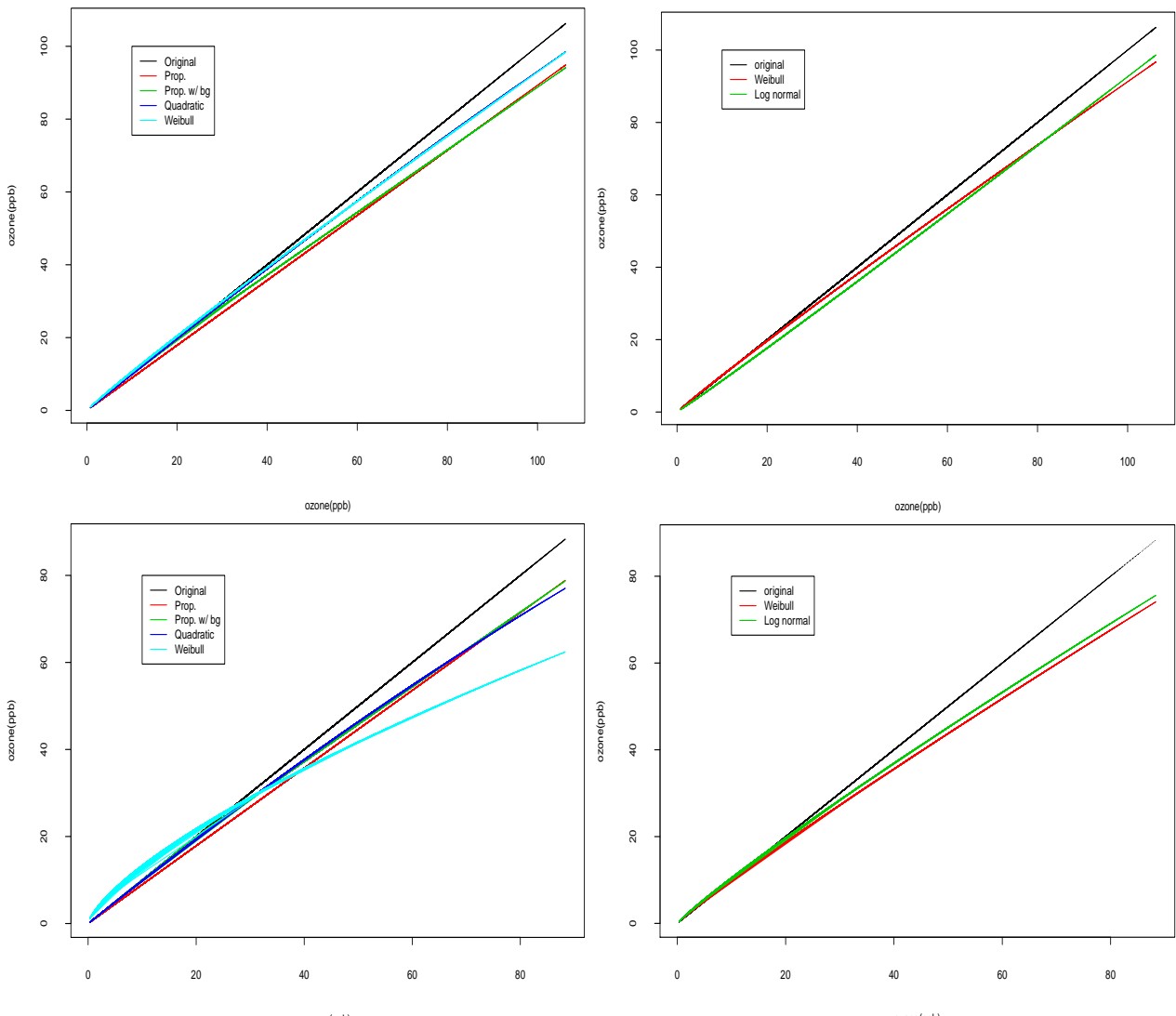

**Figure 2.** Q-Q plots based on various rallback functions (**left**) and statistical rollbacks (**right**) based on ozone data in New York (**top**) and Detroit (**bottom**) during 2003.

**Table 1.** Posterior means and 95% confidence intervals of total mortality reduction per 1000 deaths (1998–2000): common rollback.

| Model | Reg. | Prop. | Prop. w/ BG | Quadratic | Weibull | Stat.W | Stat.LN |
|---|---|---|---|---|---|---|---|
| CDL | level 75 | 1.87 | 1.09 | 0.95 | −0.62 | 0.98 | 0.94 |
| | | (1.07, 2.62) | (0.62, 1.53) | (0.54, 1.35) | (−1.18, −0.11) | (0.52, 1.37) | (0.49, 1.33) |
| | level 70 | 2.86 | 1.66 | 1.46 | 0.60 | 1.51 | 1.47 |
| | | (1.65, 4.09) | (0.95, 2.39) | (0.84, 2.08) | (0.10, 1.09) | (0.88, 2.14) | (0.83, 2.10) |
| | level 60 | 4.92 | 2.86 | 2.51 | 2.98 | 2.61 | 2.54 |
| | | (2.84, 7.02) | (1.64, 4.09) | (1.43, 3.61) | (1.53, 4.30) | (1.48, 3.75) | (1.46, 3.65) |
| UDL | level 75 | 1.91 | 1.11 | 0.98 | −0.62 | 1.03 | 0.98 |
| | | (1.15, 2.72) | (0.66, 1.59) | (0.58, 1.39) | (−1.17, −0.07) | (0.63, 1.44) | (0.59, 1.40) |
| | level 70 | 2.92 | 1.70 | 1.49 | 0.59 | 1.58 | 1.52 |
| | | (1.66, 4.14) | (0.97, 2.41) | (0.84, 2.10) | (0.10, 1.09) | (0.92, 2.18) | (0.85, 2.13) |
| | level 60 | 5.01 | 2.92 | 2.56 | 3.05 | 2.79 | 2.60 |
| | | (2.93, 7.19) | (1.66, 4.23) | (1.49, 3.70) | (1.73, 4.38) | (1.72, 3.93) | (1.56, 3.78) |

**Table 2.** Posterior means and 95% confidence intervals of total mortality reduction per 1000 deaths (1998–2000): city-specific rollback.

| Model | Reg. | Prop. | Prop. w/ BG | Quadratic | Weibull | Stat.W | Stat.LN |
|---|---|---|---|---|---|---|---|
| CDL | level 75 | 1.96 | 1.20 | 0.98 | −0.52 | 1.02 | 0.99 |
| | | (1.06, 2.92) | (0.61, 1.79) | (0.51, 1.46) | (−1.33, 0.25) | (0.54, 1.51) | (0.50, 1.48) |
| | level 70 | 2.75 | 1.66 | 1.38 | 0.44 | 1.42 | 1.39 |
| | | (1.47, 4.02) | (0.87, 2.46) | (0.74, 2.03) | (−0.36, 1.26) | (0.78, 2.09) | (0.75, 2.05) |
| | level 60 | 4.62 | 2.76 | 2.40 | 2.67 | 2.56 | 2.49 |
| | | (2.64, 6.69) | (1.58, 4.06) | (1.36, 3.52) | (1.38, 3.99) | (1.52, 3.70) | (1.45, 3.62) |
| UDL | level 75 | 1.96 | 1.19 | 0.98 | −0.51 | 1.01 | 0.99 |
| | | (0.93, 2.91) | (0.52, 1.79) | (0.45, 1.46) | (−1.36, 0.36) | (0.49, 1.49) | (0.50, 1.48) |
| | level 70 | 2.78 | 1.69 | 1.40 | 0.45 | 1.44 | 1.41 |
| | | (1.50, 3.96) | (0.88, 2.42) | (0.74, 1.99) | (−0.42, 1.23) | (0.79, 2.08) | (0.74, 2.03) |
| | level 60 | 4.81 | 2.88 | 2.50 | 2.79 | 2.62 | 2.58 |
| | | (2.66, 7.05) | (1.55, 4.26) | (1.36, 3.70) | (1.41, 4.31) | (1.49, 3.84) | (1.44, 3.79) |

**Table 3.** Posterior means and 95% confidence intervals of total mortality reduction per 1000 deaths (1998–2000): city-specific rollback and CDL model.

| Covariate | Reg. | Prop. | Prop. w/ BG | Quadratic | Weibull | Stat.W | Stat.LN |
|---|---|---|---|---|---|---|---|
| Daily Ave | level 75 | 1.96 | 1.20 | 0.98 | −0.52 | 1.02 | 0.99 |
| | | (1.06, 2.92) | (0.61, 1.79) | (0.51, 1.46) | (−1.33, 0.25) | (0.54, 1.51) | (0.50, 1.48) |
| | level 60 | 4.62 | 2.76 | 2.40 | 2.67 | 2.56 | 2.49 |
| | | (2.64, 6.69) | (1.58, 4.06) | (1.36, 3.52) | (1.38, 3.99) | (1.52, 3.70) | (1.45, 3.62) |
| Daily Max | level 75 | 2.32 | 1.91 | 1.66 | 2.48 | 2.12 | 2.09 |
| | | (1.54, 3.14) | (1.26, 2.59) | (1.09, 2.27) | (1.62, 3.40) | (1.59, 2.79) | (1.55, 2.73) |
| | level 60 | 6.19 | 5.10 | 4.25 | 5.13 | 4.87 | 4.59 |
| | | (4.11, 8.31) | (3.33, 6.87) | (2.79, 5.75) | (3.41, 6.91) | (3.38, 6.35) | (3.09, 6.06) |
| Daily 8 h Max | level 75 | 2.10 | 1.61 | 1.36 | 1.90 | 1.56 | 1.52 |
| | | (1.26, 2.92) | (0.96, 2.23) | (0.79, 1.94) | (1.13, 2.67) | (0.96, 2.15) | (0.94, 2.12) |
| | level 60 | 5.53 | 4.23 | 3.43 | 4.34 | 3.67 | 3.55 |
| | | (3.37, 7.50) | (2.52, 5.76) | (1.99, 4.69) | (2.52, 5.91) | (2.26, 4.94) | (2.14, 4.81) |

**Table 4.** Posterior means and 95% confidence intervals of total mortality reduction per 1000 deaths (1998–2000): city-specific rollback and CDL model.

| Covariate | Reg. | Prop. | Prop. w/ BG | Quadratic | Weibull | Stat.W | Stat.LN |
|-----------|------|-------|-------------|-----------|---------|--------|---------|
| Bayesian | level 75 | 1.96 | 1.20 | 0.98 | −0.52 | 1.02 | 0.99 |
| | | (1.06, 2.92) | (0.61, 1.79) | (0.51, 1.46) | (−1.33, 0.25) | (0.54, 1.51) | (0.50, 1.48) |
| | level 60 | 4.62 | 2.76 | 2.40 | 2.67 | 2.56 | 2.49 |
| | | (2.64, 6.69) | (1.58, 4.06) | (1.36, 3.52) | (1.38, 3.99) | (1.52, 3.70) | (1.45, 3.62) |
| MLE | level 75 | 2.23 | 1.31 | 1.09 | −0.30 | 1.21 | 1.19 |
| | | (1.14, 3.21) | (0.62, 1.91) | (0.54, 1.58) | (−1.15, 0.48) | (0.66, 1.70) | (0.64, 1.69) |
| | level 60 | 5.07 | 2.93 | 2.52 | 3.03 | 2.56 | 2.52 |
| | | (3.00, 7.28) | (1.65, 4.29) | (1.42, 3.68) | (1.69, 4.37) | (1.49, 3.73) | (1.39, 3.69) |
| Pooled MLE | level 75 | 1.92 | 1.16 | 0.95 | −0.50 | 1.15 | 1.10 |
| | | (1.79, 2.05) | (1.09, 1.24) | (0.89, 1.02) | (−0.61, −0.38) | (1.08, 1.22) | (1.04, 1.18) |
| | level 60 | 4.65 | 2.78 | 2.40 | 2.69 | 2.49 | 2.44 |
| | | (4.37, 4.92) | (2.62, 2.93) | (2.27, 2.54) | (2.48, 2.88) | (2.38, 2.62) | (2.32, 2.59) |

**Table 5.** Posterior means and 95% confidence intervals of total mortality reduction per 1000 Deaths (1998–2000): without high temperature (>85).

| | Prop. | Prop. w/ BG | Quadratic | Weibull | Stat.W | Stat.LN |
|-----------|-------|-------------|-----------|---------|--------|---------|
| With High Temp | 1.96 | 1.20 | 0.98 | −0.52 | 1.02 | 0.99 |
| | (1.06, 2.92) | (0.61, 1.79) | (0.51, 1.46) | (−1.33, 0.25) | (0.54, 1.51) | (0.50, 1.48) |
| Without High Temp | 1.39 | 0.83 | 0.67 | −0.47 | 0.75 | 0.71 |
| | (0.33, 2.41) | (0.15, 1.46) | (0.15, 1.17) | (−1.36, 0.45) | (0.22, 1.26) | (0.18, 1.22) |

## 5. Concluding Remarks

As the rollback approach provides an adjusted ozone process in a deterministic manner, we introduce a statistical rollback approach based on the quantile matching method to allow for uncertainty and thus provide flexible rollback sets. The proposed methods are applied to epidemiologic data (NMMAPs) and we assess the impact of new ozone regulation standards on public health under various settings. Another possible model is to consider different time lags in the Poisson process model with constrained or unconstrained distributed lags together. Using respiratory deaths rather than nonaccidental deaths may also provide useful results. One of the issues that need to be addressed is collinearity between ozone and temperature, for example. Other studies revealed strong effects in the Northeast and Industrial Midwest, less strong but still significant effects in the Southeast and possibly Southern California. Thus, we can also consider a regional or spatial structure in the method. Finally, the parametric rollback approach using quantile matching may allow uncertainty in ozone process itself, which can be extended to a fully Bayesian framework.

**Author Contributions:** Y.K. concieved the idea and developed the method presented herein and wrote the paper. J.L. performed data analysis. Both authors have read and approved the final manuscript. All authors have read and agreed to the published version of the manuscript.

**Funding:** This research was supported by Basic Science Research Program through the National Research Foundation of Korea (NRF) funded by the Ministry of Education (No. 2018R1D1A1B07043352).

**Acknowledgments:** We are grateful to the Editor-in-chief, Associate Editor and the anonymous referees for their helpful comments.

**Conflicts of Interest:** There are no potential conflict of interest.

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
