# Peer review of "A Risk Assessment for Ozone Regulation Based on Statistical Rollback"

_applsci, doi:10.3390/app11052388_

Round 1
Reviewer 1 Report
This manuscript presents an interesting study on technical and statistical approaches to assess the relationship between ozone regulation and human mortality. The authors use a rollback method applied to epidemiological data to study the effects of ozone regulations on human health and they adopt the quantile matching method to allow for uncertainty in the rollback procedure and thus provide flexible rollback sets. The analysis performed by the authors is accurate and the methodology used is appropriate to the study.
Point 1: Abstract. The purpose and the scientific novelty of the study is stated.
Pont 2: Introduction: Although the introduction describe the state of the art of the scientific problem it’s need to introduce more bibliography.
Point3: section 2. (2.2) I propose to describe more clearly this section and all symbols used.
Point 4 lines 122 /123 There is a small typo in the lines: “Association”
Point 5 section 4: figure 2 improve the text of the legend and the axes.
Author Response
We are grateful for the valuable comments from you. We consider your comments carefully, and make a correction of our paper or an answer for their questions. The attached are our responses to your comments.

Reviewer 2 Report
The paper is suitable for publication after the explaining the abreviations used in paper
Author Response

(The authors gave the same response as above.)
